# Hyperglycemia and Oxidative Stress: An Integral, Updated and Critical Overview of Their Metabolic Interconnections

**DOI:** 10.3390/ijms24119352

**Published:** 2023-05-27

**Authors:** Patricia González, Pedro Lozano, Gaspar Ros, Francisco Solano

**Affiliations:** 1Department of Biochemistry, Molecular Biology B and Immunology, Faculty of Chemistry, Campus de Espinardo, University of Murcia, 30100 Murcia, Spain; patricia.gonzalez@fresenius-kabi.com (P.G.); plozanor@um.es (P.L.); 2Fresenius Kabi España, S.A.U., Marina 16-18, 08005 Barcelona, Spain; 3Department of Food Technology, Nutrition and Bromatology, School of Veterinary, Campus de Espinardo, University of Murcia, 30100 Murcia, Spain; gros@um.es; 4Department of Biochemistry, Molecular Biology B and Immunology, School of Medicine-IMIB, Campus Ciencias de la Salud, University of Murcia, Avda. Buenavista 32, El Palmar, 30120 Murcia, Spain

**Keywords:** hyperglycemia, oxidative stress, ROS, protein glycation, AGE & RAGE, NFκB, Nrf2, pro-oxidant and antioxidant enzymes, insulin resistance, diabetes type II

## Abstract

This review focuses on the multiple and reciprocal relationships that exist between oxidative stress, hyperglycemia and diabetes and related metabolic disorders. Human metabolism uses most of the consumed glucose under aerobic conditions. Oxygen is needed in the mitochondria to obtain energy, as well as for the action of microsomal oxidases and cytosolic pro-oxidant enzymes. This relentlessly generates a certain amount of reactive oxygen species (ROS). Although ROS are intracellular signals necessary for some physiological processes, their accumulation leads to oxidative stress, hyperglycemia, and progressive resistance to insulin. A cellular pro-oxidant versus an antioxidant equilibrium would regulate ROS levels, but oxidative stress, hyperglycemia, and pro-inflammatory conditions feed back to each other and the relevance of the interconnections tends to increase those conditions. Hyperglycemia promotes collateral glucose metabolism through protein kinase C, polyols and hexosamine routes. In addition, it also facilitates spontaneous glucose auto-oxidation and the formation of advanced glycation end products (AGEs), which in turn interact with their receptors (RAGE). The mentioned processes undermine cellular structures, finally giving place to a progressively greater degree of oxidative stress with further hyperglycemia, metabolic alterations, and diabetes complications. NFκB is the major transcription factor involved in the expression of most of the pro-oxidant mediators, while Nrf2 is the major transcription factor regulating the antioxidant response. FoxO is also involved in the equilibrium, but its role is controversial. This review summarizes the key factors linking the diverse glucose metabolic routes enhanced in hyperglycemia with ROS formation and vice versa, emphasizing the role of the major transcription factors involved in the desirable balance between pro-oxidant and antioxidant proteins.

## 1. Introduction: Oxidative Stress and Aerobic Metabolism

The human metabolism comprises many usually enzyme-catalyzed biochemical reactions occurring inside cells, or in extracellular physiological fluids, although not only catalysis but the transport of metabolites and other actors could be considered. The metabolic reactions allow the correct functionality of organs and the whole body, they constitute the molecular basis of life and are one of the main hallmarks of either health [1] or disease if disturbances in these reactions occur [2]. Metabolism makes possible all the cellular activities (i.e., energy production from nutrients, maintaining the structural integrity of cells and tissues, the response to external and internal stimuli, growth, cell division and so on). The major catabolic routes are responsible for the nutrient transformation into intermediate metabolites that are basically used (i) to obtain energy needed for anabolic processes, muscular contraction and changes in membrane potential (ii) to build units of structural macromolecules (structural polysaccharides, phosphor, glycolipids, proteins and nucleic acids), and (iii) to form modulators and mediators for specific physiological functions (hormones, eicosanoids, chemokines, neurotransmitters and others).

From an ancient evolutionary point of view, the appearance of oxygen (O_2_) in the Earth atmosphere symbolized an outstanding advantage for the energetic metabolism of aerobic organisms. The amount of energy that can be obtained from nutrients is much higher in aerobiosis than that in anaerobiosis. As early as in 1783, Antoine-Laurent de Lavoisier emphasized the use of oxygen in the respiration of living animals: “Respiration is simply a slow combustion of carbon and oxygen, similar in all respects to what occurs in a burning lamp or candle, and from this point of view, respiring animals are really combustible bodies that burn and consume themselves” [3].

Animal cells use glucose as the main source of energy, and glycolysis as the main degradative route to obtain that energy. Aerobic glycolysis leads to pyruvate, the final product of the route for AcetylCoA (AcCoA) synthesis inside the mitochondria. AcCoA feeds the Krebs cycle to produce energy through the reduction of reduced nicotinamide adenine dinucleotide (NADH) and flavin adenine dinucleotide (FADH_2_) cofactors, which transfer its energy to oxidative phosphorylation ATP synthesis by means of the electron transport chain (ETC). According to basic biochemical concepts, these processes couple glucose and oxygen as perhaps the two simple most essential molecules for the maintenance of the energetic metabolism of animals [4].

Although the use of oxygen in metabolism is a good deal for cells in energetic terms, it has a relatively high cost from a sustainable point of view. The intracellular aerobic oxidation of glucose and other biomolecules to produce carbon dioxide and water is not totally clean. Molecular oxygen, even in the most stable oxygen triplet form, has two unpaired electrons, and this determines the easy appearance of reactive oxygen species (ROS) as water-alternative partially reduced subproducts. ROS evolve from molecular oxygen into water according to the monoelectronic transferences shown in Figure 1. Superoxide (O_2_^●−^) and hydrogen peroxide (H_2_O_2_) are the most common ROS, as they are normal products of so many enzymes and metabolic reactions. However, other more reactive and dangerous reactive species such as the hydroxyl radical (^●^OH), singlet dioxygen and RNS (reactive nitrogenous species) are also formed under appropriate conditions.

In addition to water, the appearance of ROS is a collateral minor process even in healthy mitochondria, which cannot be fully eliminated. Its formation is inherent to the use of ETC and oxygen as an electron acceptor. Tiny amounts of superoxide anion are generated in the mitochondria and cytosol, and H_2_O_2_ is formed by many oxidases and reactions of the secondary metabolism. Superoxide is transformed into hydrogen peroxide by SOD. In addition, both ROS react with each other due to the Haber–Weiss and Fenton spontaneous reactions to produce very hazardous ROS, such as hydroxyl radicals (^●^OH) or singlet oxygen, with high deleterious and mutagenic potential. Harman first proposed the free radical theory [5] to connect these reactive species with aging-related diseases. Although this theory cannot completely explain the aging process, the role of oxidative damage in the tissue decline in its functional course with aging is widely accepted.

Despite the potential damaging effects usually associated to ROS, the concept that any ROS is harmful for cells and that any antioxidant is beneficial is just a myth [6]. Under physiological concentrations, the generation of low amounts of ROS is a normal process, as those species are signal molecules necessary to regulate normal metabolism, cell growth, differentiation, apoptosis, and autophagy.

Cells are subjected to a redox balance necessary for metabolic homeostasis, a sort of equilibrium between pro-oxidant and antioxidant signals. When an aberrant production of ROS exceeds the buffering capacity of the antioxidant defense systems, or when antioxidant enzymes are defective, ROS accumulation occurs, and oxidative stress appears. Pathological levels of ROS may lead to cellular dysfunction, cell death, and finally whole organ failure. Oxygen overload, mitochondrial dysfunction, elevated levels of glucose free fatty acids, and other external or genetic factors can increase ROS formation to give place to oxidative stress [7,8].

In specific cells, the redox balance should be broken to obey specific physiological functions. For instance, in the innate immune system, macrophages, granulocytes, and dendritic cells possess biochemical mechanisms for inducing a burst of ROS for bacterial killing [9]. In addition to this defense oxidative function, ROS are potent inflammatory initiators that play essential roles in the inflammatory response by provoking the synthesis and secretion of numerous pro-inflammatory molecules, such as prostaglandins (PGE2 and others), leukotrienes (LTB4), or cytokines (TNFα, IL-1β, IL-6) [10,11]. Oxidative stress is often related to inflammation and inflammasome activation as all these biomolecules promote autocrine and paracrine effects on the surrounding cells and these cells respond either to ROS signals or ROS generation.

## 2. Glucose Metabolism, Hyperglycemia, Oxidative Stress, and Pathological Effects

Glucose comes with a cost in terms of oxidative stress. From a chemical point of view, glucose is an aldohexose, and thus it is a moderately reactive molecule. The modulation of glycemia and the regulation of glucose uptake by cells, are essential points for health. It is well-known that hyperglycemia causes metabolic disorders since it triggers “aberrant” pathways that promote oxidative stress in human tissues [12,13,14]. Recent reviews emphasizing the close association between hyperglycemia, oxidative stress and inflammation have been published in a number of human metabolic dysfunctions and pathologies related to the liver [15,16], adipose tissue [17], skeletal muscle [18,19], kidney [20,21], cardiovascular system [22,23,24], retina [25], osteocartilaginous system [26], neurodegenerative processes [27,28] and transient diseases such as pre-eclampsia during pregnancy [29].

Among the metabolic alterations and pathologies promoted by hyperglycemia in the organs and tissues referred to above in the mentioned order, the liver shows the non-alcoholic fatty liver disease [30] characterized by increased oxidative stress, inflammation, the development of insulin resistance and fat accumulation in the hepatic tissue. The adipose tissue is metabolically active, containing macrophages and stromal cells in addition to adipocytes. These additional cells produce adipokines to regulate carbohydrate metabolism and its sensitivity to insulin that finally favors inflammation and hyperglycemia processes, which involve an interconnection between dyslipidemia, obesity, diabetes mellitus (DM), oxidative stress, and inflammation [31]. The main adipokines related to the insulin action in peripheral tissues are leptin, resistin or adiponectin, which participate in appetite regulation [32,33].

Oxidative stress and hyperglycemia have been also associated with muscular frailty due to the imbalance of muscular proteostasis, and the accelerated decay of the functional capacity and strength of myocytes [34]. In addition to having a direct effect on mature myocytes, it has been recently shown that high glucose concentrations impede the proliferation of satellite cells, which are muscle-specific stem cells [19]. In this way, hyperglycemia could actively erode pre-existing myocytes and the regenerative capability of skeletal muscle myofibers.

The kidney is a continually active mitochondria-rich cell organ, which makes it vulnerable to damage caused by hyperglycemia-induced oxidative stress. It is known that this state can accelerate chronic kidney disease. Patients at advanced stages of chronic kidney disease show oxidative stress, which is associated with hypertension and other disorders. Thus, various antioxidant pharmacological agents (e.g., N-acetylcysteine, and vitamins C and E) directed at reducing oxidative stress are used as potential therapeutical agents for adult chronic kidney disease patients [20,35]

Endothelial dysfunction is the main alteration involved in the pathogenesis of macrovascular vessels and subsequent cardiovascular diseases, from atherosclerotic to heart failure. This dysfunction is mostly related to the increase in the expression of cell-surface adhesion molecules in both endothelial and blood immune cells causing a breakage of the redox serum balance, ROS accumulation, oxidative stress, and hyperglycemia [9,36]. Concerning the retina, which is also a very oxidative tissue, oxidative stress is a critical contributor to the pathogenesis of diabetic retinopathy, which is the most usual microvascular complication of diabetes [25,37].

Altogether, it is currently clear that the relationships among hyperglycemia, oxidative stress and pro-inflammatory signals are closely interconnected in numerous tissues [38,39]. In turn, all the above-mentioned pathological conditions involve a high percentage of the worldwide population, especially older adults (>60 years old), and a comprehension of the molecular biochemical mechanisms is essential for the advance of novel therapeutical approaches to ameliorate or neutralize harmful effects.

Osteoarthritis (OA) is a degenerative and painful joint disease caused by the wear, or breakdown, of the cartilage covering the ends of bone joints due to chronic inflammation. It leads to localized pain and limited joint movement [40,41]. Healthy normal chondrocytes maintain a balance between the generation and degradation of collagen, proteoglycan, and other components of the extracellular matrix (ECM) [42]. During OA progression, the components of the cartilage are degraded matrix metalloproteinases (MMPs) and some specific serine proteases including high-temperature requirement A (HtrA), whose activities are enhanced in chondrocytes and synoviocytes subject to oxidative stress [43,44]. A loss of the ECM balance within the joint increases the ROS content, impairs the glucose metabolism in those cells and induces inflammation.

The aim of this review is to describe in depth the major molecular aspects of the hyperglycemia–oxidative stress–inflammation interconnections, as an enabling tool for better understanding the interactions of three vertices of this vicious triangle. The collation of the most recent data with classical concepts related to glucose metabolism tissue-specific molecular mediators, regulation of the ROS level, or pro-inflammatory signaling molecules should be considered a keystone accounting for the molecular events occurring in human tissues, and for the development of novel and coordinated therapies to ameliorate the associated pathological and clinical consequences of hyperglycemia as one of the major causes of oxidative stress and vice versa.

## 3. Metabolic Links between Hyperglycemia and Oxidative Stress at Molecular Level

As stated above, ROS formation is inherent to oxygen consumption during mitochondrial respiration and the action of oxidases. However, ROS accumulation until to the occurrence of authentic oxidative stress is really a consequence of a variety of factors. Most of them are usually related to physiological and biochemical processes that involve aberrant glucose metabolism due to hyperglycemia.

Glucose has a variable trajectory in human tissues, including both metabolic and spontaneous transformations (Figure 1). The main pathway for glucose metabolism is glycolysis (the green central vertical route), which supplies energy as ATP in most of the human tissues. Human glycolysis can occur in aerobiotic or anaerobiotic lactic fermentation. The first option is preferred as it is much more energetic than lactate formation. Pyruvate is imported to the mitochondrial matrix to form AcCoA and feeds the Krebs cycle, ETC and oxidative phosphorylation [4]. 

Other usual pathways for glucose metabolism are the pentose phosphate, glucuronate and glycogen pathways (in green in Figure 1, on the left). Their contribution to glucose metabolism depends on the tissue, cellular energetic demand, and amount of glucose, but in general these routes do not have significant harmful effects in terms of oxidative stress due to ROS generation. However, glucose can also undergo other transformations (salmon-colored, on the right), that are quantitatively less important in relation to glycolysis, but are enhanced by hyperglycemia and closely related to ROS generation. Furthermore, spontaneous glucose reactions are also responsible for important links between hyperglycemia and oxidative stress. In turn, their occurrence feeds back the hyperglycemic conditions, causes low-to-moderate-grade inflammation and induces a slow and concomitant advance from transient to chronic hyperglycemia.

The above-mentioned pathways are summarized in Table 1. They seem to be separate ways to generate ROS, but they are interconnected with each other (Figure 2). Most of the mechanisms converge to produce intracellular signals leading to an exacerbation of oxidative stress and the appearance of molecular mediators related to pathological effects.

### 3.1. Cell Respiration and Mitochondrial Generation of Superoxide and Other ROS

The major form of glucose consumption is the aerobic glycolysis of glucose to pyruvate and subsequent intramitochondrial AcCoA to produce the reduced cofactors NADH and FADH_2_ in the Krebs cycle. Those cofactors download electrons in mitochondrial complexes I and II, respectively. The electronic flow through mitochondrial components of the ETC is not 100% bi-electronic, and mono-electronic transferences give place to unpaired electrons, resulting in the formation of coenzyme Q (CoQ) semiquinonic species and phospholipid oxidation [45]. ROS production is related to the electronic leakage from the ETC. The higher the number of electrons retained in ETC complexes, higher the production of superoxide anion.

Around 2% of the oxygen consumed in the ETC generates superoxide in healthy mitochondria, but the amount can increase by at least twice the original amount in dysfunctional mitochondria due to damage in mitochondrial complexes and lipid peroxidation. Thus, aerobic glycolysis could become an important source of superoxide, depending on the integrity of mitochondrial ETC machinery [46].

The intramitochondrial matrix contains an important arsenal of antioxidant defense systems to maintain a low superoxide level [47,48]. There are two intramitochondrial superoxide dismutase (SOD) isoenzymes able to transform the generated superoxide in H_2_O_2_, which is then reduced to water by some peroxiredoxins [49]. However, the Haber–Weiss and Fenton reactions lead to the formation of tiny amounts of hydroxyl radicals and singlet oxygen, and overall mitochondrial ROS production cannot be totally extinguished. Those species slowly but continuously induce lipid peroxidation and damage mitochondrial structures, including mitochondrial and DNA complexes [50,51,52]. This cumulative damage leads to the dysfunction of the organelle, making respiration less efficient in ATP synthesis, and ROS formation increases as long the machinery of mitochondrial impairment maintains it. Consequently, mitochondrial defense to minimize ROS is not enough and a certain amount of hydrogen peroxide is released into the cytosol, contributing to an increase in cellular oxidative stress. Hyperglycemia increases glucose consumption through this route and consequently accelerates mitochondrial dysfunction.

The cytosolic part of glycolysis also provides electrons that are stored in NADH, so the higher the level of hyperglycemia, the higher the amount of NADH produced. After NADH shuttling into mitochondria, complex I is overcharged by NADH, and the more electrons are downloaded, the more superoxide is produced [47]. Accordingly, the importance of mitochondrial impairment in provoking metabolic disturbances in cell cultures subject to a high-glucose media and diabetes-induced animal models is well-documented. For instance, it has been proven that an oversupply of NADH in diabetes impairs mitochondrial function, enhances cellular oxidative stress, and increases cell death [53]. Furthermore, short-time exposure of mice to a high-glucose diet does not produce mitochondrial dysfunction, but prolonged treatment (>1 month) induces a diabetic state in parallel to an altered mitochondrial structure and the dysfunction of the skeletal muscle. Conversely, it is proven that ROS production and mitochondrial dysfunction in the skeletal muscle is implicated in the development of diabetes mellitus type 2 (DMT2) [54], but glycemia normalization decreases ROS production and partially restored mitochondrial integrity in myocytes, pointing out the link between hyperglycemia, mitochondrial dysfunction and ROS production. Similarly, a close interrelation between hyperglycemia and mitochondrial impairment as major source of ROS has been proposed in diabetic retinopathy [55] and in the dysfunction of pancreatic beta cells in diabetic animals [56].

### 3.2. Collateral Pathways of Glucose Metabolism Related to Hyperglycemia

Although the data referred to above illustrate the relevant role of mitochondrial respiration in ROS generation in hyperglycemia, several experimental data indicate that the appearance of oxidative stress is really the result of several confluent mechanisms [49,57]. In addition to the mitochondrial ETC, there are three other intracellular metabolic routes located in the cytosol and ER for using glucose that are directly related to ROS generation and can contribute to oxidative stress. They are called aberrant or collateral glucose metabolism [58]. The amount of glucose consumed through these routes under normal glycemic conditions is low, but obviously hyperglycemia facilitates those reactions. The three routes are regulated by specific enzymes (Figure 1) and are tissue-dependent according to the expression of such enzymes. In the retina and other tissues, the polyol route can metabolize up to 30% of the available glucose in hyperglycemia.

Glycolysis and collateral routes are not independent routes, and there are crosstalk mechanisms among them. For instance, it was observed that the inhibition of the ETC in bovine endothelial cells prevented not only ROS generation in hte mitochondria, but also other events such as glucose-induced activation of PKC, the formation of AGEs and NFκB activation [55,57]. Nevertheless, for simplicity, they are described separately.

#### 3.2.1. Protein Kinase C Route

PKCs are a large family of enzymes with serine/threonine kinase activity found in membrane-bound and soluble forms of all cellular types. They are engaged in several signal transduction pathways essential in the cellular response to nutrients and hormonal stimuli. Some of these PKC types are activated by hyperglycemia through different mechanisms [59,60,61]. The most direct one is related to the glycolysis route, as the higher consumption of glucose through this route implies a higher availability of the glycolytic intermediate glyceraldehyde-3-phosphate. This triose can be reduced to glycerol-3-phosphate for the synthesis of DAG (Figure 1, lower right), a strong activator of PKC acting as second messenger. Although the net increase in DAG might be low, small increases could produce deep metabolic disturbances due to the amplification of the transduction signals involving PKC. The activation of PKC isoforms triggers several cellular responses such as the expression of various growth factors, but PKC-dependent NADPH oxidase (NOX) activation and the AGE–RAGE pathway are considered the two major routes for high glucose-induced ROS overproduction [62]. PKC activation and the AGE–RAGE pathways (Section 3.3.2) are two inseparable events associated to hyperglycemia. In addition, PKC causes an increase in the activity of some NOX isoenzymes responsible for the DMT2 caused by the dysfunction and inhibition of the insulin secretion of pancreatic beta cells [63].

One example of the pathological complications of hyperglycemia and the associated occurrence of oxidative stress is PKC-dependent vascular calcification in the endothelial tissue of diabetic patients [61]. Vascular calcification is characterized by the hardening of the medial layer of the artery wall caused by the deposition of hydroxyapatite minerals into the extracellular matrix. In this system, PKC and AGE–RAGE are activated, and the corresponding target in the endothelial cells is the activation of the NOX1 isoform, as well as concomitant SOD inhibition. These changes induce the Runt-related transcription factor 2 (RunX2) that enhances the expression of alkaline phosphatase and other bone matrix proteins, switching the phenotype of the vascular smooth muscle cells to that of osteoblast-like cells and hydroxyapatite deposition.

According to other studies, PKC activation stimulates the activity of other pro-oxidant enzymes, such as nitric oxide synthase (NOS), xanthine oxidase (XO), and lipoxygenases (see Section 4.2 on), all of them leading to an increase in ROS generation [63]. It is widely accepted that ROS and PKC activation mutually establish a vicious crosstalk during hyperglycemia-induced atherosclerosis. However, direct use of PKC inhibitors as therapeutical agents is still pending, as PKC isoforms show complex regulation and a variety of crosstalks with other signal transduction systems which may exert antagonistic effects on the redox balance [64,65]. For instance, some PKC isoforms show two different domains with Cys-rich motifs [66] that can be involved in cysteine thiol-based redox signaling that regulates Nrf2 and thioredoxin active forms (Section 5 and Section 5.1.4). The N-terminal domains can be oxidized by peroxides for PKC activation, but the C-terminal domain can be reduced by an antioxidant producing PKC inhibition [13,67].

#### 3.2.2. Polyol Route

In normoglycemia, intracellular glucose is predominantly phosphorylated into glucose 6-phosphate and enters the glycolytic pathway. However, in hyperglycemia, there is a remarkable increase in the polyol pathway, accounting for up to 30% of intracellular glucose [68,69,70]. Therefore, this is usually the most important collateral pathway for glucose metabolism in hyperglycemia with inactive glucokinase, and thus historically, this is a very well-studied mechanism for the appearance of oxidative stress and the subsequent diabetic complications.

The route consists of just two reactions. The first one, the rate-limiting step, is the reduction of glucose into sorbitol catalyzed by aldose reductase, using NADPH as a cofactor. Then, the polyol can be re-oxidized to fructose by sorbitol dehydrogenase, using NAD^+^ as a cofactor. Importantly, both reactions are specific to the mentioned pyridine nucleotide cofactors [68,71]. Therefore, the two consecutive reactions decrease the cytosolic NADPH/NADP^+^ ratio but increase that of NADH/NAD^+^ (Figure 1 upper left), and NADPH is replaced by NADH. NADH induces oxidative stress via the generation of ROS through the mitochondrial ETC, as described below. Moreover, NADPH depletion leads to a decrease in the GSH/GSSG ratio, as the reduction of oxidized glutathione (GSSG) by glutathione reductase requires NADPH as a cofactor [72]. The lower availability of reduced GSH also contributes to oxidative stress. Diabetic rats compensate for oxidative stress by increasing the phosphate pentose pathway to increase the NADPH level [73]. This compensatory mechanism supports the importance of NADPH in the maintenance of a low level of ROS and the importance of the polyol pathway in promote higher levels.

Concerning the formation of glucose-derived monosaccharides, the complete polyol pathway converts glucose into fructose, and an overproduction of fructose can lead to harmful metabolic consequences. On one hand, fructose can chemically glycate proteins, leading to protein dysfunction (Section 3.3.1). On the other hand, it is known that fructose could be further metabolized to produce 3-deoxyglucose and fructose-3-phosphate, both of which are potent nonenzymatic glycation agents [74]. Thus, the flux of glucose through the polyol pathway would increase AGE formation.

The pathway leads to an excessive accumulation of intracellular ROS in other tissues of diabetic patients including the heart, vasculature, neurons, eyes, and kidneys [75]. Aldose reductase has been isolated from several human and animal tissues including the eye, liver, placenta, kidney, erythrocyte, brain, and the heart and skeletal muscles [70]. Moreover, NADPH content was also found to be lower in diabetic lungs and pancreases, and the decrease was quantitated as around 15% under the diabetic lens [76].

In turn, certain tissues such as the retina or some renal cells show low sorbitol dehydrogenase activity, so that sorbitol accumulates instead of fructose. Under these conditions, NADH is not overproduced, but it has been proposed that sorbitol accumulation could trigger cellular osmotic stress, one underlying mechanism contributing to diabetic retinopathy or nephropathy [70].

All these data strongly suggest that the polyol pathway contributes in several ways to increase oxidative stress and plays an important role in the pathogenesis of diabetes in human patients. Indeed, a number of aldose reductase inhibitors are currently being investigated to prevent diabetic complications such as nephropathy, retinopathy, cardiomyopathy, and neuropathy [57,69,70,77], but this point is beyond the scope of this review.

#### 3.2.3. Hexosamine Route

Like the previous two collateral pathways of the glucose metabolism, the hexosamine biosynthetic route also contributes to the onset of hyperglycemia and vice versa. It is well-known that the hexosamine pathway is activated in diabetes, and reciprocally the activation of this route is associated to the development of insulin resistance [78].

This pathway is an early stage in glycolysis and begins with an amination reaction that converts the glycolytic intermediate fructose 6-phosphate to glucosamine 6-phosphate, catalyzed by rate-limiting glutamine-fructose-6-phosphate amido transferase activity (Figure 1, middle right). After other enzyme-catalyzed reactions, the final key metabolite of this pathway is UDP-N-acetylglucosamine (UDP-NAcGlcNH_2_), an UDP-activated derivative which is able to bind to proteins via O-linked glycosylation. This modification alters the structure and functionality of many proteins [79]. Most of these modifications occur in the ER, giving place to ER stress characterized by the dysfunction of chaperones and other ER proteins. It is believed that this mechanism constitutes a sort of nutrient sensor to monitor glucose availability according to the O-glycosylation degree of those target ER-resident proteins. According to this, it has been demonstrated that the increased expression of amido transferase activity in pancreatic β-cells triggers ER stress [80] resulting in β-cell dysfunctions, characterized by a higher generation of cytosolic hydrogen peroxide and a concomitant decrease in GLUT2 transporter, glucokinase and insulin production [81,82].

In addition, the hexosamine route activates other cellular signals and transcription factors able to change gene expression patterns [83]. Some of the upregulated genes have been identified, including those encoding for transforming growth factor beta (TGFβ) [82] and for the inhibitor of the plasminogen-1 activator (PAI-1) [84]. Both proteins are related to the vascular and renal effects of hyperglycemia. TGFβ expression has fibrogenic effects and increases renal fibrosis in the arterial walls and renal mesangeal cells (Section 4.1). In addition, this factor contributes to redox imbalance by increasing ROS production and through the suppression of the expression of antioxidant enzymes. In turn, ROS induce TGFβ expression, contributing to the vicious self-stimulatory cycle between oxidative stress and hyperglycemia [85].

To conclude on the metabolic disturbances related to the abuse of the hexosamine route, some results suggest the formation of minor amounts of glucosamine itself (GlcNH_2_), derived from the route by dephosphorylation, which might also contribute to the onset and progression of diabetes [81]. This non-activated hexosamine competitively inhibits glucose uptake by pancreatic β-cells or hepatocytes, lowering insulin secretion or increasing insulin resistance, respectively. However, recent data indicate that the contribution of this mechanism to ER stress and fibrosis is uncertain. The importance of all the hexosamine-derived modifications in cellular proteins and metabolic disturbances related to diabetic complications has been recently reviewed [82,86].

### 3.3. Spontaneous Glucose Reactions

#### 3.3.1. Protein Glycation

The higher importance of the spontaneous reactions of glucose due to hyperglycemia (upper right in Figure 1) is directly related to the pathological consequences of hyperglycemia. Free glucose dissolved in blood and physiological fluids is majorly found as a stable D-glucopyranose form, but according to the chemical equilibrium, around 1% is found as a linear aldohexose. The aldehyde group is susceptible to undergoing spontaneous chemical reactions with other dissolved biomolecules. Thus, this group slowly reacts with amine groups forming a Schiff base to place an Amadori conjugate between glucose with blood proteins [87]. This addition is also a Maillard-type reaction, but in the hyperglycemic context, is called spontaneous protein glycation to be distinguished from intracellular enzyme-catalyzed protein glycosylation occurring in the ER. The reaction is also possible with other serum monosaccharides such as fructose or galactose, but by far glucose is the most abundant one, especially under hyperglycemic conditions. Glycated proteins usually change its native structure and subsequently its functional properties [88].

One well-known example of this reaction is the formation of glycated hemoglobin (mostly the HbA1c isoform), which has been used in clinical biochemistry as a routine long-term hyperglycemic marker for decades. Glucose can be bound to hemoglobin through the N-terminal residue of the protein and also some side chain ε-amino groups of lysine residues, and the extent of glycation depends on the time and the level of glycemia. The formation of hemoglobin–glucose adducts decreases the affinity of the protein to 2,3-biphosphoglycerate, dysregulates the iron-binding site, the efficiency of oxygen transport and finally the red blood cell metabolism and lifespan [89]. Obviously, other plasmatic proteins can undergo glycation. Special attention is currently paid to the glycation of human serum albumin, the most abundant one in blood [90,91] and the effects of the accumulation of glycated albumin in the transport properties of this protein. In turn, glycation can also inactivate specific antioxidant enzymes, as observed with the glycation of the extracellular SOD3 isoenzyme (Section 5.1), impairing serum antioxidant defenses against oxidative stress [92].

#### 3.3.2. Glucose Auto-Oxidative Lysis; AGE Formation and AGE–RAGE Interactions

Although the direct glycation of hemoglobin by glucose is used as a clinical marker of persistent hyperglycemia, the effect of this glycation on the level of oxidative stress in plasma seems to be negligible. With regard to this, it is much more relevant to other types of spontaneous free glucose reactions. Minor amounts of plasmatic free glucose undergo diol-tautomerization, transforming the aldose form into the enediol form. The enediol form is more reactive and easily reacts with traces of oxidized Fe(III) or Cu(II) ions present in the blood, so that the 6C chain is fragmented. The reaction yields short-chain ketoaldehydes, mostly methylglyoxal or glyoxal, as well as superoxide, which directly contributes to oxidative stress [93,94,95]. Furthermore, short-chain ketoaldehydes are much more reactive than unaltered glucose is and they rapidly react with the amine groups of proteins. These species give place to an ill-defined unordered and progressive degradation of the involved proteins with the concomitant appearance of a series of products generically called AGE (advanced glycation end products) [96,97,98,99,100]. Interestingly, the Amadori complexes derived from glucose, fructose or fructose-derived protein glycation undergo further slow reactions, eventually leading to the formation of AGEs too. These reactions are ill-defined, but AGE formation has been known for decades and used as a marker of plasma oxidative stress due to hyperglycemia. AGEs are related to oxidative stress, but also inflammatory responses, changes in gene expression patterns and cellular damage involved in the pathogenesis of hyperglycemia and almost all tissue-specific diabetic complications. AGEs are a heterogeneous group of compounds, and some of them resemble fragments of several amino acid residues, including lysine and arginine (i.e., pentosidine) but also histidine methionine or cysteine coming from ketoaldehyde-induced protein breakage [101].

The number of AGEs is still growing, and they are involved in hundreds of complex reactions with other biomolecules [88]. Hence, AGE formation directly alters cellular structures or promotes the formation of cross-links between molecules of different tissues. On this point, the basement membrane of the ECM is regularly active. Remarkably, in addition to those direct effects, AGEs bind with relative affinity and specificity to a number of membrane proteins called receptors of advanced glycation end products (RAGE). RAGEs are diverse in tissular distribution, although most of them are integrins or belong to the immunoglobulin superfamily [102]. They are found in different cell types, but stand out in immune, vascular epithelial or renal mesangial cells [103]. The interaction of AGE–RAGE (Figure 3) can activate several transduction signals, some kinases (PKC, JAK, p38-MAPK, and ERK), GTPases, TGFβ, transcription factors (such as NFκB, Section 4.1), and some pro-oxidant enzymes such as NOX and NOS (Section 4.2.1 and Section 4.2.2). All these effects usually lead to a strong exacerbation of intracellular oxidative stress and the stimulation of the collateral pathways of glucose metabolism that are described above (PKC activation, polyol and hexosamine routes) completing a positive feedback circle in relation to the oxidative stress signals transmitted by the AGE–RAGE initial interaction. Consequently, AGEs are also able to release pro-inflammatory cytokines and to induce the expression of pathological endothelial vascular proteins. In summary, AGE–RAGE interaction is an important molecular event responsible for the development of diabetic complications in different tissues, including retinopathy, nephropathy, neuropathy, and macro/microvascular damage. More specific physiological details about these processes can be found in recent reviews [28,61,95].

In summary, keto groups in the unaltered glucose molecule or short-chain ketoaldehydes coming from glucose-fragmented oxidation give place to spontaneous chemical reactions, from protein glycation to AGEs. These reactive molecules, directly or through binding to RAGEs, induce progressive moderate to severe harmful effects on several tissues [100]. The rate and importance of the effects increase as much as the hyperglycemia level rises, and in chronic hyperglycemia those reactions become one of the major causes of the correlation between oxidative stress and inflammation and of the clinical complications found in diabetic patients.

## 4. The AGE–RAGE Pathway and the NFκB Activation

### 4.1. Main Molecular Mediators Related to Hyperglycemic Pathological Effects

The transcription factor called nuclear factor-κB (NFκB) modulates gene expression in diverse cellular processes such as stress responses to a variety of noxious stimuli. This factor was proposed as the specific major cellular sensor of oxidative stress [104], but more recent studies indicate that the activation of the NFκB pathway depends on the duration and context of exposure to the stimuli as well as the cell type. This transcription factor is involved in the cellular response to several stress stimuli, it is always associated to the AGE–RAGE signals, and it is also involved in the intracellular transmission of oxidative stress [105,106] and the activation of the inflammatory response [107]. It is confirmed that when AGEs bind to RAGEs or the cellular production of ROS overwhelms the antioxidant capacity of cells, the activation of NFκB activates the transcription of some pro-oxidant enzymes as well as many important proteins related to the side effects of oxidative stress. As a result, recent studies have studied the implications of the transcription factor NFκB in the development of metabolic disorders including DMT2, as well as most of the pathological effects associated to the disease [22,106,108].

One example illustrative of the involvement of NFκB in the appearance of pathological complications is the development of endothelial dysfunction in vascular vessels, one of the clinical symptoms found in diabetic patients. The vascular endothelium contains a single layer of cells oriented to the lumen of blood vessels. These cells are responsible for the maintenance of vascular homeostasis, including the modulation of vascular tone, maintenance of blood fluidity, immune response, and vessel permeability. Nitric oxide (NO) is the major substance responsible for the maintenance of the vascular homeostasis. Under normal physiological conditions, NO shows vasodilatory and anti-thrombotic effects. This molecule is an arginine-derived gas that inhibits the expression of NFκB and therefore lowers the secretion of proinflammatory cytokines, vasoconstrictors, and prothrombotic adhesion molecules [109,110].

Oxidative stress causes endothelial inflammation and dysfunction. This is an activated, pro-inflammatory and pro-coagulant state characterized by the exacerbated expression of the cell-surface adhesion molecules required for the recruitment and attachment of leukocytes to the activated platelet, the expression of pro-inflammatory cytokines and the appearance of pro-thrombotic factors [9,36,111,112]. The advent AGE and its binding to the RAGE at the endothelial cell membrane promotes the activation of NFκB and the expression of a set of molecules that dramatically change the features of the endothelium and neutralize the NO vasodilatory and anti-adhesion effects.

NO secretion decreases either via an inhibition of endothelial nitric oxide synthase (eNOS) expression [113] and/or impaired eNOS activity caused by tetrahydrobiopterin (BH_4_) uncoupling or its limited availability [114] (Section 4.2.2). As a second alternative, the impairment is accompanied by ROS formation through peroxynitrite, and oxidative stress is reinforced [110]. LDL is also oxidized by the AGE–RAGE pathway and the oxidized lipoprotein may react with NO to further reduce its bioavailability [115]. The decrease in the amount of the vasodilator (NO) is probably the main cause of the onset of endothelial dysfunction, as well as the increase in the level of vasoconstrictor factors (such as ET-1 and Angiotensin II). Moreover, NFκB orchestrates the appearance of other molecular mediators that enable cellular adhesion (such as intercellular and vascular cell adhesion molecule 1 (ICAM-1 and VCAM) and, monocyte chemoattractant protein-1 (MCP-1)), as well as prothrombotic and pro-coagulant effects (such as plasminogen activator inhibitor-1 (PAI-1, which counteracts the physiological anti-coagulant tissue plasminogen activator (tPA)) [112,116]. According to those molecular events, it has been observed that hyperglycemia gives place to a lower secretion of NO in the vasculature of diabetic patients and that impaired NO production also contributes to the generation of the oxidative stress responsible for glomerular renal damage.

As mentioned above, another transduction signal activated by the AGE–RAGE pathway is the expression of TGF𝛽, which also has a role in the onset of diabetic renal and vascular complications [117,118,119]. TGF𝛽 triggers the phosphorylation of a family of specific transcriptional modulators called Smads (the name stands for ‘suppressor of mother against decapentaplegic’) [120] as well as other AGE–RAGE-activated kinases (i.e., protein38-mitogen activated protein kinase and extracellular-signal regulated kinase p38, MAPK and ERK).

The TGF𝛽 signal leads to an increase in the expression of type I collagen [121] and other ECM proteins that accelerate fibrogenic growth [118]. Fibrosis is one of characteristic complications associated to chronic hyperglycemia in the vascular system and renal tissues. The vascular smooth muscle cells responsible for the tractable and contractile characteristics of the artery walls, but higher collagen secretion alters their morphology and properties, increasing the rigidity and stiffness of the walls [106]. Concerning the kidney, it has been demonstrated that TGFβ promotes alterations in diabetic nephropathy, including ECM growth, fibrosis, thickening of glomerular and capillary basement membranes and proliferation of mesangial cells. Inhibition of TGFβ expression with natural agents, such as artemisinin, has been recently proposed as a novel appropriate approach for attenuating TGFβ-mediated pathological effects due to oxidative stress in the kidney [122].

### 4.2. Enzymatic Pro-Oxidant Systems Directly Involved in ROS Generation

The mechanisms for regulating ROS levels are directly related to the activity of several enzymes that maintain the intracellular redox balance. There are two large groups, the pro-oxidant enzymes, generally oxidases, and antioxidant enzymes, which act as ROS scavengers. Both groups respond to distinct stimuli and obey distinct functions, both contribute to the redox equilibrium, and the imbalance in these activities due to the predominance of the first group is mostly responsible of the appearance of the oxidative stress [123]. The activity of most of them is related to the level of glucose since NFκB and other transcription factors also exert their regulatory pro-oxidant action by changing the expression of the genes encoding such enzymes.

Hyperglycemia can induce the activity of up to eight types of oxidases through direct and indirect mechanisms that finally generate oxidative stress. Some of those oxidases are very closely related to the AGE–RAGE pathway and collateral glucose metabolism, although the link and their significance are tissue-specific. The reactions catalyzed by these oxidases are summarized in Table 1.

#### 4.2.1. NADPH Oxidase (NOX)

These oxidases are mostly ubiquitous in all human tissues. They catalyze the formation of superoxide from oxygen using NADPH as a specific electron donor [9,36], so they are likely the major and a more versatile cellular source of ROS. There are seven isoforms usually bound to the cytosolic membrane with non-identical tissular distribution [49] to fulfil a variety of roles. All the isoforms are complex oligomers containing a guanine nucleotide-binding subunit and another oxidase subunit usually named phagocyte oxidases as most of them are characterized as belonging to phagocytic cells to produce high amounts of ROS for the oxidative degradation of the engulfed molecules or pathogens. In those phagocytic cells, NOXs are powerful non-counterbalanced pro-oxidant enzymes adapted to the role of those cells.

NOX1 and NOX6 are encoded by mitochondrial DNA and are associated to complex I of the ETC [124]. In non-phagocytic cells, NOX activity shows apparently opposite effects related to different roles and a dual action [125]. On one hand, NOX activity causes NADPH depletion. On the other hand, a NOX-catalyzed reaction is a direct source of superoxide and subsequently other ROS when acting in parallel to SOD.

Focusing on NADPH, this cofactor is a double-edged sword in terms of the cellular redox balance. NADPH is a cytosolic reductant when it acts as a substrate of glutathione reductase, but it becomes a source of superoxide when it acts as a NOX substrate. NADPH acts as the key and rate-limiting reductant antioxidant agent to maintain low H_2_O_2_ levels through the glutathione recycling coupled with the glutathione peroxidase/glutathione reductase (GPx/GR) system [126,127]. Oppositely, NOX produces NADPH depletion and consequently a decrease in the cellular glutathione level. In this regard, NOX gives place to an effect such as that of the aldose reductase involved in the polyol pathway (Section 3.2.2).

Hence, NADPH levels are a key factor for the redox equilibrium. The main source of cytosolic NADPH is the phosphorylated pentose pathway (Figure 1, on the left), which in the liver supplies around 60% of the total cytosolic NADPH. Thus, NADPH formation is dependent on glucose metabolism and this route is also important for other redox processes although ROS are not directly involved. It is not surprising that diabetic rats compensate for ROS stress through increasing the phosphate pentose pathway to increase the NADPH level [73]. On the other hand, note that the key enzyme of NADPH formation is glucose-6-phosphate dehydrogenase, and the importance of this route in the maintenance of the redox state is illustrated in erythrocytes where the inactivation of this enzyme leads to hemolytic anemia due to the deficit in NADPH [128,129].

Bearing in mind the antioxidant role of NADPH, it is proposed that under normal conditions the role of NOX is the generation of moderate amounts of ROS necessary for the activation of certain transduction signals beneficial to cell metabolism, but NAPDH should not be completely depleted, so that a significant amount of the cofactor can be used for other processes, such as the maintenance of the GPx/GR system and the appropriate level of reduced glutathione. Recently, it has been demonstrated that genetic deletion of NOX4 enhances formation of solid tumors [130]. NOX4 knockout mice show angiotensin II-mediated vascular inflammation and massive endothelial dysfunction [131], indicating the importance of this isoenzyme in endothelial cells. According to this, under healthy conditions, NOX4 exerts protective effects in those cells. Moderate levels of superoxide anion production and SOD-derived H_2_O_2_ are required for normal redox signaling which promotes vasodilation and vascular remodeling, stimulating appropriate angiogenesis, migration, and cellular proliferation. In peripheral cells, low levels of NOX are needed to enhance the insulin response via the inhibition of the phosphatase TENsin homolog (PTEN) and other phosphatases prolonging the action of the insulin activated phosphatidyl inositol 3 kinase/”Ak strain” transforming the protein (also named protein kinase B) (PI3k/Akt) signal [132]. Thus, it is clearly proven that NOX activity is essential for normal cell functions and appropriate metabolic responses.

On the other hand, under pathological conditions, as far as NOX activity would be stimulated and the redox equilibrium would be broken, this activity would produce considerable amounts of ROS and authentic oxidative stress due to both, the formation of superoxide and the depletion of NADPH. For instance, hyperglycemia inactivates adenosine monophosphate kinase (AMPK) and this is a strong signal for NOX activation [49,133].

NOX4 is the most abundant isoform in most cell types [9]. The overstimulation of the NOX4 isoform is related to the complications in the diabetic retina [134], endothelial vasculature [36], and kidney [135] and the appearance of insulin resistance in peripheral tissues [132]. For instance, one of the more harmful effects occurs in the kidneys, as AMPK inactivation stimulates the NOX4 isoform, and the superoxide burst damages glomerular cells, contributing to diabetic nephropathy [135,136].

Elevated levels of NOX4 were also found to correlate with the progression of vascular dysfunction. High NOX4 activity gives place to excessive oxidative stress that induces an increase in the levels of pro-inflammatory cytokines as well as promoting the expression of NFκB and a subsequently higher amount of adhesion molecules including (ICAM-1, VCAM, MCP-1 and E-selectin). These molecules, in coordination with the decrease in or impairment of NO formation (Section 4.2.2) induce the conversion of healthy endothelium into a pro-thrombogenic phenotype.

In addition to NOX4, the induction of the NOX1, NOX2 and NOX5 isoforms has also been found to promote inflammation and endothelial dysfunction in animals with experimentally induced diabetes [137]. The main roles of higher expressions of NOX1 and NOX5 seem to be related to ROS production under pathophysiological conditions, such as the vascular calcification related to the activation of the PKC and AGE–RAGE pathways [62]. Augmentation of NOX2 activity is responsible for the ROS accumulation leading to insulin resistance and alterations in the translocation of GLUT4 transporters observed in the skeletal muscle [132]. The NOX5 isoform is the only NADPH oxidase activity modulated by intracellular Ca^2+^ levels. Increased expression levels of NOX5 were found in arteries of patients with coronary artery disease [9,36].

#### 4.2.2. Uncoupled Nitric Oxide Synthase (NOS)

NOSs are a family of three enzymes that catalyze the formation of NO via arginine oxidation [9]. Nitric oxide is a vasodilator gas regulating blood pressure and platelet aggregation and is the major substance responsible for the maintenance of vascular homeostasis. The enzyme is found in constitutive and induced isoforms and is mostly expressed in epithelial cells (eNOS) and neurons. Under healthy conditions, the coupled physiological enzyme that forms NO is a dimer that needs tetrahydrobiopterin (BH_4_) as a cofactor [13,36,138]. Remarkably, hyperglycemia causes a deficit in this cofactor, as BH_4_ is oxidized to 7,8-dihydrobiopterin (BH_2_). Under reduced BH_4_ availability, the eNOS dimer is partially uncoupled into the monomeric form [114] which leads to enzymatic dysfunction producing a low amount of superoxide in addition to NO (Table 1), so the impaired monomeric eNOS becomes a source of ROS. Importantly, superoxide reacts spontaneously and rapidly with NO to form peroxynitrite. This is a very powerful RNS oxidant agent, which is more harmful than ROS, and it can oxidize several biomolecules, including glutathione, and the available BH_4_. In addition, it is also able to promote the nitration of tyrosine residues of proteins, generating devastating oxidative stress and causing the activation of the intracellular AGE–RAGE pathway and NFκB expression, resulting in an irreversible injury of mitochondria and cell apoptosis [52]. These uncoupling NOS-derived events, in parallel with NOX4 activation described in the Section 4.2.1, contribute to endothelial cell dysfunction [139,140] and renal damage [138] related to diabetes-associated cardiovascular complications and nephropathy.

According to this, it has been proven that BH_4_-based treatment has the potential to ameliorate the complications of diabetes in different tissues. BH_4_ treatment leads to a partial improvement in diabetic patients [138]. For instance, the augmentation of BH_4_ biosynthesis in hyperglycemic human aortic endothelial cells via the gene transfer of GTP cyclohydrolase I (GTPCH, the rate-limiting enzyme for de novo BH_4_ synthesis) rescued eNOS dimerization and clean NO synthetic activity [141]. Furthermore, the administration of BH_4_ to diabetic mice ameliorated glucose intolerance and insulin resistance [114]. Recent data confirm that chronic treatment with BH_4_ ameliorates the cardiorenal effects of diabetes [142].

#### 4.2.3. Cyclooxygenase (COX)/Prostaglandin G/H Synthase (PGHS)

This family of two/three enzymes is involved in the formation of pro-inflammatory eicosanoid mediators (prostacyclins, prostaglandins and tromboxanes). They catalyze the rate-limiting step of the pathway and constitute one of the molecular links between oxidative stress and inflammation. The reaction uses molecular oxygen as a substrate to form a series of (endo)peroxides and ROS as subproducts. The two historical and recent denominations are due to the partial or total description of the mechanism of action, as the enzymes catalyzes two consecutive activities, oxidase and peroxidase, transforming arachidonate released from cell membrane into unstable intermediate PGG2 (cyclooxygenase activity) and then to another endoperoxide PGH2 (peroxidase activity) [143]. Therefore, COX is the classical and most widely used name, but PGHS is more precise and recommended [144]. Endoperoxides release ROS as by-products of the reaction, and also act as signal mediators, so that pathway is activated by surrounding oxidative stress and it can contribute to the maintenance of a high ROS level during the inflammatory response. There are two human isoenzymes (COX1, or PGH1, and COX2, or PGH2), although the gene encoding COX1 may give place to a third isoenzyme, COX3. Those isoenzymes display differential sensitivity to aspirin and many widely used non-steroid anti-inflammatory drugs (NSAID). The first one is involved in the synthesis of tromboxanes and the second one is related to prostaglandin synthesis in the acute phase of inflammation, which is more closely related to fever and pain.

Hyperglycemia influences this activity, although the mechanisms are still partially unclear. On one hand, it was reported that hyperglycemia enhances the inflammatory response by upregulating COX2 in monocytes, which would be relevant in terms of the vascular complications associated to diabetes [145]. According to that, isolated peripheral blood monocytes from diabetic patients show higher levels of COX2 mRNA in comparison to those from normal volunteers. Moreover, the induction of COX2 activity by high glucose concentrations is blocked by inhibitors of the mitochondrial ETC, hexosamine metabolism, NOX, PKC, p38 mitogen-activated protein kinase and NFκΒ, indicating the multiple interactions from the induction of that enzyme and several pathways or signaling systems associated to hyperglycemia. On the other hand, it has also been reported that the overexpression of COX2 protects cultured hepatocytes from the increased hepatic apoptosis observed in experimental models of diabetes, and the addition of PGE2 mimics the protective effect observed with COX2 [146]. Other studies describe that hyperglycemia causes a slow but stimulant effect on PGHS1, and the most recent data confirm the complexity and tissue-dependent interconnections between hyperglycemia and the use of inhibitory NSAID drugs [147]. Further investigations are underway with regard to this because of the high worldwide prevalence of both, diabetes and the use of NSAID drugs.

#### 4.2.4. Lipoxygenases (LPOs)

This is another heterogenous family of Fe-oxidases related with inflammatory processes and production of ROS, similarly to PGHS. They catalyze the peroxidation of unsaturated fatty acids at the cellular membranes to synthesize hydroperoxy and epoxy derivatives that evolve to lipoxins, leukotrienes and other signal molecules, with the parallel by-production of ROS due to the instability of those oxygenated species [12,148]. There are up to 13 isoforms differing in the preferential substrate and the position of the oxygen attack to yield the peroxide or epoxy derivate [149]. In relation to hyperglycemia, high-glucose content stimulates LPO coupled to other pro-oxidant enzymes. The LPO isoforms are cell- and tissue-dependent. They seem to be involved in diabetic retinopathy [150,151]. At this stage, it is important to remind the ω3 polyunsaturated fatty acids peroxidation is associated to a 3C-fragments breakage of the hydrocarbon chain, releasing as malondialdehyde (MDA), used as global LPO-dependent and independent lipid peroxidation marker. MDA and other aldehydes are reactive, and they can become an extra source of additional ROS [152].

#### 4.2.5. Xanthin Oxidase (XO)

This is a key enzyme involved in purine catabolism that catalyzes the two-steps or one-step oxidation of hypoxanthine or xanthine to uric acid. XO is a complex protein containing molybdenum, Fe-S centers, pterin and FAD as cofactors. Like eNOS, this enzyme shows a ROS-devoid clean dehydrogenase activity, but under oxidative stress conditions, a switch from the reductase form to the oxidase form occur and the enzyme catalyzes the reduction of molecular oxygen into both superoxide and H_2_O_2_ [153]. This oxidase form reinforces the oxidative stress in polymorphonuclear, endothelial, epithelial cells, and fibroblasts, especially under the renal damage [154] or vascular impairment observed in diabetes [155]. As in other cases of increased pro-oxidant activity, this activity positively feeds back cytokine release, inflammation, and oxidative damage [156]. According to that, increased XO level was found in plasma of diabetic patients, and recently allopurinol was found to inhibit the XO activity and decrease the superoxide level in diabetic patients and animal models [157,158].

#### 4.2.6. Heme Oxygenase (HO1)

The heme catabolic pathway generates pro-oxidant and antioxidant compounds, and consequently, can influence cellular redox balance. On one hand, heme releases iron and porphyrins generate photochemical reactions. Iron could be involved in deleterious reactions, as this metal facilitate the Fenton reaction and the appearance of hydroxyl radical. Thus, heme degradation shows potential toxicity in cellular metabolism. However, the activation of HO1 is regulated by Nrf2 (Section 5.1) and so it is considered part of the ubiquitous cellular antioxidant response to oxidative stress [159]. This could be justified by the antioxidant properties of another product of the HO1-catalyzed reaction, biliverdin, and the subsequent metabolite of the route, bilirubin [160].

HO1 seems to be relevant in endothelial cells, although maximal activity is found in the spleen. Interestingly, the enhancement of this activity stimulates insulin secretion, and it can be accompanied by a reduction in the severity of hyperglycemia. Hemin, an activator of HO1, improved glucose tolerance, reduced insulin resistance, and ameliorate the inability of insulin to increase the number of GLUT4 at the cell membrane, the protein required for glucose uptake [161]. On the other hand, hyperglycemia increases the HO1 activity and decreases the erythrocyte half-life. These opposite effects indicate the dual role of this activity as pro and antioxidant, although the prominent role seems to be dependent on insulin availability.

#### 4.2.7. Myeloperoxidase (MPO)

This is a particular peroxidase with important antimicrobial functions expressed in cells of the immune system (macrophages, monocytes, and neutrophils). This enzyme shows a higher redox potential as compared to other peroxidases. As peroxidase, it uses hydrogen peroxide as substrate for oxidizing chloride into hypochlorite, a potent and bleaching oxidant species related to ROS that confers these cells high capability to degrade microbial pathogen components. According to that, it is considered a pro-oxidant rather than an antioxidant enzyme. In addition, myeloperoxidase accomplishes NOX for the generation of a high oxidizing media. Both activities play important roles in the killing function of neutrophils by the availability of a NOX-O_2_^●−−^-H_2_O_2_-MPO-HOCl highly oxidizing system. MPO activity is higher in diabetic neutrophils [162] but the phagocytic and killing functions of neutrophil phagocytosis are impaired. However, hypochlorite formed by infiltrated neutrophils enhances the oxidative stress and plays important roles in the vascular injury, amplifying the endothelial dysfunction associated to diabetes [163]. The enzyme is also involved in the oxidation of serum lipoproteins and components of the endothelial cells, as well a reduction in NO and loss of cell viability in hyperglycemia [164]. Finally, it has been recently proposed that MPO may be involved in a mechanism of protein carbamylation potentially involved in the formation of atherosclerotic plaques [165].

#### 4.2.8. Cytochrome P450 (CYP)

This is most diverse and versatile family of oxidases occurring in all tissues and involved in many biochemical processes in the secondary metabolism or in detoxification reactions. Thus, they are involved in the hydroxylation of several bioactive molecules (prostaglandins, bile acids, corticoids, and cholecalciferol) as well xenobiotic metabolism (ethanol, toxins, and drugs). All CYP isoenzymes are hemoproteins located in mitochondrial and microsomal fraction. Superoxide is one of the main byproducts of catalysis, but other ROS can also be formed. Their ubiquity and diversity make it difficult to establish links between CYP isoenzymes and hyperglycemia. However, CYP can also contribute to the ROS-induced generation of hyperglycemia, and on the other hand, diabetes influences the levels of activity for some CYP isoenzymes [13]. For instance, hyperglycemia strongly induces the CYP2E1 mitochondrial isoform [166]. A few other CYP isoenzymes have been found to be stimulated in diabetic patients, either altering the catabolism of CYP-substrate drugs (i.e., paracetamol [167] or the synthetic pathways of bioactive molecules needed for regulating physiological process [168,169].

## 5. Main Transcription Factors Involved in the Antioxidant Response: Nrf2 and FoxO

Oppositely to the above-described mechanisms linking hyperglycemia to the enhancement of oxidative stress and vice versa, an imbalance in intracellular redox homeostasis can also trigger antioxidant responses to counteract the harmful and self-stimulatory effects. The most powerful response is regulated by some transcription factors that change in the gene expression patterns in an opposite way to that in NFκB. The central role of this antioxidant response is fulfilled by the transcription factor termed nuclear factor (erythroid-derived 2)-like 2 (Nrf2). Many studies [170,171] have proposed that Nrf2 is the master regulator of the antioxidant response, coordinating the expression of proteins contributing to the anti-inflammatory response with antioxidant enzymes to lower ROS levels (Table 2). Although it is out of the scope of this review, it is worth mentioning that Nrf2 is also a key factor in the treatment of several types of cancer as the activation of the antioxidant defense results in chemoresistance, inactivating drug-mediated oxidative stress and protecting cancer cells from drug-induced cell death. Thus, the modulation of Nrf2 should be considered so as not to hinder treatments aimed at killing cancer cells by stimulating oxidative stress. More information can be found in recent reviews related to ovarian, cervical, endometrial, and prostate cancers [172,173,174].

The activation of Nrf2 is part of reactive cysteine thiol-based redox signaling [66]. An excellent and illustrative updated review on the molecular mechanism of Nrf2 activation has been recently published [170]. It is called the Kelch-like ECH Associated Protein 1(KEAP1)-Nrf2-Antioxidant Response Element (ARE) signaling pathway. Briefly, KEAP1 is an E3 ubiquitin ligase tightly bound to cytosolic Nrf2 in normal redox homeostasis which inactivates its action via the ubiquitination and proteasome-dependent degradation of this protein. In response to oxidative stress, a sensor mechanism related to the Cyst thiol-based redox signaling triggered via the oxidation by 4 Cys residues of KEAP1 allows Nrf2 to escape ubiquitination and translocate to the nuclei, where it promotes its antioxidant transcription program by binding to ARE (antioxidant response element) regions of DNA. Nrf2 increases the expression of an ample set of genes encoding proteins related to antioxidant defense. The set of antioxidants enzymes upregulated by Nrf2 are superoxide dismutase (Cu/ZnSOD), peroxiredoxins, thioredoxins, glutathione peroxidase/glutathione reductase as well as other proteins participating in GSH detoxification, the production of glutathione-S-transferase (GST)- and GSH-generating proteins, GCLC and GCLM (glutamate cysteine ligase catalytic and regulatory subunits [175,176,177] and heme oxygenase 1 [HO1) [31]. The regulation is so broad and coordinated that not only those antioxidant enzymes, but even some enzymes of the phosphate pentose pathway are also stimulated to contribute to NADPH production and at the same time increase the consumption of glucose via a non-generating ROS pathway. Other antioxidant enzymes, such as catalase and some SOD isoenzymes, are not directly regulated by the Nrf2/ARE pathway, but it has been reported that those activities also contribute to the antioxidant response to oxidative stress ROS in an Nrf2-independent fashion [178].

Another mechanism contributing to decrease in ROS levels is the direct scavenging of species with diverse antioxidant biomolecules (vitamins A and E, polyphenols, flavonoids, chalcones, sulforaphane, and curcumin) [175,179]. These antioxidants are micronutrients that could be ingested from foods, and they are well-known useful agents that prevent hyperglycemia-induced oxidative stress [180]. For instance, α-tocopherol (vitamin E) is an excellent mitochondrial antioxidant protecting lipid peroxidation on the mitochondrial membrane due to hydrophobicity. However, the topic is actively investigated, and new and more efficient molecules are being characterized [31]. Some of these antioxidants are not only direct ROS scavengers but are also able to induce strong activation of the Nrf2 signaling pathway [181]. Up to four distinct ways to stimulate the Nrf2 signaling pathway have been proposed, including the inhibition of Keap1, modification of the upstream mediators of Nrf2, improvement of the nuclear translocation of Nrf2 and finally, enhancement of the expression of Nrf2 and its target genes [182]. Although a complete review of this issue is beyond the aims of this review, one illustrative example is curcumin, which has been proposed as an effective therapeutical treatment for diabetes mellitus and chronic renal disease.

The contribution of Nrf2 to preventing oxidative stress in hyperglycemia has been demonstrated in several models, such as renal function and physical exercise models. Thus, the activation of the Nrf2/ARE signal attenuates hyperglycemia-induced damage in podocytes [183]. Both podocytes from diabetic mice and patients show more severe mitochondrial dysfunction, higher levels of urinary albumin and creatinine and a lower expression of the proteins synaptopodin and nephrin than those of podocytes affected by the previous activation of the Nrf2/ARE pathway. In addition, the activation of the Nrf2/ARE pathway partially protects against mesangial matrix deposition, collagen deposition and mitochondrial structural damage. The mechanism is partially related to sirtuin 1 activation via increasing NAD^+^ levels (Section 5.1.6), suggesting the multiple interactions of the Nrf2/ARE antioxidant response with other pathways. In a different model, a skeletal muscle model of rats, it was described that Nrf2 contributes to the prevention of oxidative stress and insulin resistance. Both the Nrf2 and GLUT4 transporter were activated in the gastrocnemius of rats after aerobic exercise, although the response was much lower in rats showing insulin resistance [178], suggesting the involvement of this hormone.

In addition to Nrf2, other families of transcription factors are involved in the antioxidant response, such as the forkhead box, classO (FoxO) [184], named according to the presence of that structural motif. Four members are present in humans, FoxO1a, 3a, 4 and 6 [185]. They are complex regulators of the cellular stress response. Under stress stimuli, including oxidative stress, and similarly to Nrf2, FoxO is translocated to the nucleus to upregulate a series of target genes, thereby promoting an adaptive response. In the case of oxidative stress, FoxO stimulates the expression of some genes encoding for antioxidant response located in different subcellular compartments, such as mitochondrial SOD2, Prx3 and Prx5 isoenzymes, Trx2 and the corresponding Trx2 reductase, peroxisomal catalase and some plasmatic antioxidant-related proteins, such as selenoprotein P, the main serum protein involved in selenium transport to GPx [186]. According to this, diabetic nephropathy has been associated with the downregulation of FoxO in addition to the activation of NOX4 activity [187]. Moreover, downregulation of both FoxO and Sirt1 contributes to the progression of oxidative stress in endothelial cells [188].

Importantly, in addition to FoxO expression, its regulation is much more complex than that of Nrf2, and the interplay between ROS and FoxO under physiological and pathophysiological conditions can change the final cellular response. Undoubtedly, FoxO can act as a cellular redox sensor to target the antioxidant response, but chronic high ROS levels are able to produce post-translational modifications of the FoxO proteins, either via phosphorylation, acetylation or ubiquitination. These modifications alter FoxO stability and its cytoplasmic/nuclear translocation [184,189]. FoxO proteins are recruited from the nucleus to the cytoplasm and degraded via the ubiquitin–proteasome pathway so that the antioxidant response is neutralized [185]. Concerning other modifications related to the hexosamine pathway (Section 3.2.3), FoxO can be NAc-Glycosylated and this results in an increased transcription of gluconeogenic enzymes and consequently elevated glucose synthesis [190]. In summary, FoxO activation under oxidative stress cannot be enough to be part of the antioxidant response and its inactivation, on the other hand, may increase metabolic disturbances in DMT2 in some tissues. Diabetic complications such as retinopathy have been associated to elevated FoxO1a transcriptional activity [191]. FoxO1a overexpression in liver and pancreatic β-cells induced diabetes in transgenic mice due to increased hepatic glucose production combined with decreased β-cell compensation [192]. In the skeletal muscle, FoxO overexpression causes atrophy. The final practical outcome is that FoxO action is not as well-coordinated as that of Nrf2 in triggering the antioxidant response even though this transcription factor is able to stimulate some antioxidant enzymes under strict conditions. Numerous experimental data indicate that FoxO activation is related to a pro-oxidant state rather than an antioxidant one, and FoxO should not be considered an important and effective part of the antioxidant response [193].

### 5.1. Antioxidant Enzymes and Antioxidant-Related Proteins

According to the previous paragraph, under normal conditions, cells would respond to ROS-induced oxidative stress by inducing an antioxidant response regulated by key transcription factors to restore the redox balance. The antioxidant response is majorly based in enzymes able to eliminate ROS, and accessory proteins. Ingestion of supplementary biomolecules can also contribute to maintaining the balance. Table 2 shows the list of antioxidant enzymes, as well as proteins regulating cysteine-sensing systems, such as the thioredoxin/thioredoxin reductase system, and sirtuins, also involved in protection against oxidative stress via a different mechanism.

Before a general description of those proteins, it should be noted that even though ROS formation under normal conditions would trigger Nfr2 expression, there are numerous reports indicating that diabetes leads to a dysfunction of the antioxidant response [194,195]. For instance, there is accumulated experimental evidence indicating that the activity of key enzymes of the antioxidant response (such as SOD or GPx) is decreased in plasma, liver, retina, or vascular endothelial cells of diabetic patients.

The balance between antioxidant and pro-oxidant signals is complex and it is subject to several signal transduction pathways. The controversial role of FoxO factors in the antioxidant response has been discussed above. Oxidative injury occurs when ROS production exceeds the buffering capacity of the antioxidant response. For instance, hyperglycemia activates NFĸB, TGFβ, pro-inflammatory and pro-oxidant signals, and the self-stimulatory action of all those signals could overwhelm the antioxidant response. For instance, hyperglycemia provokes the glycation and subsequent inactivation of some antioxidant enzymes, such as SOD [92]. In turn, the polyol pathway produces a significant decrease in NADPH, so glutathione metabolism is impaired, decreasing GR and GPx activities.

Probing the predominant action of the NFκB pro-oxidant over the Nrf2 antioxidant action, cells pretreated with inhibitors of NFĸB partly blocked hyperglycemia-induced oxidative stress. AMPK inhibitors also exacerbated ROS production, confirming the complementary role of AMPK on the antioxidant defense system [196]. This kinase is the key signal of a poor-energy cellular state that minimizes the NFκB response but stimulates thioredoxins, Nrf2 nuclear translocation and FoxO3 expression [196].

#### 5.1.1. Superoxide Dismutases (SODs)

SODs are enzymes that catalyze the conversion of superoxide into H_2_O_2_. They do not completely scavenge ROS signals as they release H_2_O_2_ as the final product of their enzymatic action. There are three different isoforms of SOD: Cu-ZnSOD1 occurring in the cytosol and mitochondrial matrix, an exclusively mitochondrial Mn-SOD2 and an extracellular Cu-ZnSOD3 [197]. The two intramitochondrial SOD isoenzymes are necessary for managing most of the superoxide generated in the ETC via conversion into H_2_O_2_. Those isoenzymes are responsible for the elimination of mitochondrial superoxide in all types of cells. When the generation of superoxide by ETC is moderate, moderate amounts of H_2_O_2_ are also formed, that are used for activating signal transduction pathways with cellular beneficial effects, such as the migration and angiogenesis of endothelial cells. However, under conditions of excess in mitochondrial superoxide due to ETC impairment, such as diabetes and other metabolic disorders, a decrease in mitochondrial SOD activity is observed. This has been detected in the photoreceptors of patients with diabetic retinopathy. Under those conditions, photoreceptors are rich in damaged mitochondria, are major contributors to oxidative stress and thus related to disease progression [198].

Many studies have investigated the antioxidant effects of SOD3 in inflammatory diseases. The determination of this extracellular isoenzyme is easier than that of cytosolic or mitochondrial SOD. Thus, it has been proposed that the increase of SOD3 protein could be a viable therapeutic approach to correct redox imbalance in inflammatory diseases such as skin, autoimmune, lung, and cardiovascular dysfunction [199] as well as in diabetic retinopathy [200].

#### 5.1.2. Catalases and Peroxidases (CAT and PO)

These two families of enzymes use H_2_O_2_ as a substrate. Note that SOD transforms superoxide into hydrogen peroxide, so other enzymes are needed for modulating the level of hydrogen peroxide using this ROS as a substrate. Most data suggest that catalase activity has not a relevant role in the antioxidant response, as the expression of the catalase-encoding gene is not enhanced by Nrf2. Mammalian catalase is exclusively located in the peroxisomes of certain tissues, mostly the liver, lung and kidney [201]. This enzyme seems to be designed for controlling the H_2_O_2_ level generated because of peroxisomal metabolism, but not for decomposing the H_2_O_2_ formed in the cytosol or that coming from the SOD action. Nevertheless, catalase may be induced in some situations as part of a non-dependent Nrf2 response to oxidative stimuli. For instance, an increase in catalase activity has been observed to increase in parallel with HO1 activity [202].

Rather than catalase-dependent peroxide decomposition, cytosolic H_2_O_2_ is usually used by peroxidases for specific tissue-dependent functions. Myeloperoxidase, thyroid peroxidase, lactoperoxidase or GPx are illustrative examples of the different roles of peroxidases in different H_2_O_2_ usages. The most ubiquitous use, which is usually involved in the maintenance of the redox balance, is glutathione peroxidase (GPx). This is a selenium protein that catalyzes the glutathione oxidation associated with H_2_O_2_ reduction (Table 2). Oxidized glutathione is subsequently converted into the reduced form via coupling with glutathione reductase (GR) [126,127]. There are four GPx isoenzymes in mammalian cells, but the most abundant is the GPx1 isoform. SOD1, GPx1, is found in the mitochondria and cytoplasm of cells [203] and is induced by Nrf2 to protect cells from oxidative stress. Accordingly, GPx1 knockdown causes cells to be more highly susceptible to death under oxidative stress. GPx1 deficiency in endothelial cells results in a decrease in NO formation but an increase in cytokine secretion and leukocyte adhesion properties [204]. Recently, it was reported that the loss of the antioxidant/oxidant balance in the serum of diabetic nephropathy patients was correlated with a low level of GPx activity [205].

#### 5.1.3. Peroxiredoxins (Prx)

This is a family of mammalian antioxidant enzymes whose function is the elimination of H_2_O_2_, too, but the mechanism of action is different from that of CAT and PO. Prx is implicated in anti-oxidative and anti-inflammatory effects in many cell types, as reviewed in [206]. They contain reduced Cys residues in the active form, but they are oxidized during the catalytic cycle and other reductant cofactors are needed for recycling these enzymes into the active form. Although mammalian cells show up to six Prx isoforms [207], there are just three types of Prx with different grades of aggregation or cofactor requirement. Prx1-5 isoforms contain two Cys residues that are oxidized into a disulfide bridge for H_2_O_2_ decomposition. Prx6 has only one Cys residue at the catalytic active site which is oxidized into sulfenic acid by H_2_O_2_ during the catalytic cycle, as a disulfide bridge cannot be formed [208]. Recycling back-oxidized Prx to thiol groups requires another coupled reductant, ascorbate, glutathione or thioredoxin [209]. Some recent data indicate that Prx3 is a mitochondrial important isoenzyme. Overexpression of Prx3 significantly diminishes mitochondrially oxidative damage. A decrease in the expression of Prx3, but not in that of Prx5, has been reported in the heart of diabetic rats, and also in high-glucose cultured cardiac cells [210]. Prx6 is also a key regulator of the cellular redox balance, with the peculiar ability to neutralize peroxides, peroxynitrite, and hydroperoxides. The absence of Prx6 causes both reduced insulin secretion and increased insulin resistance [211]. PRX6(−/−) mice develops phenotype like that of DMT2, and the stimulation of this isoform has been proposed as a target for therapeutic strategies in diabetes care.

#### 5.1.4. Thioredoxin (Trx) and Related Proteins (Trx Reductase and TxNPI)

Thioredoxins are thiol-oxidoreductases acting as major regulators of thiol-sensing cellular redox signaling. They are a family of small-sized proteins induced via the Nrf2–ARE pathway with the capability of reducing both disulfide bridges and sulfenic acids to native Cys residues of several targets, including some antioxidant enzymes such as Prx [55]. However, high concentrations of peroxide were found to induce Trx degradation [212]. Like the SOD Prx, Trxs isoforms are found in the cytosol and mitochondria. In turn, they are also able to migrate to the cell nucleus interacting with several transcription factors, including Nrf2. Complementary, oxidized Trx is reduced back into active Trx by thioredoxin reductase [213], a NADPH-dependent enzyme that contributes to completing the Trx redox cycle in a similar way to how glutathione completes the GPx/GR cycle. Note that GSH and Trx are two important reductant agents that decrease cellular oxidative stress, and both are dependent on NAPDH via the corresponding coupling with two specific reductases.

Another mechanism for modulating Trx action is mediated by the thioredoxin binding protein (TxNIP), a member of the α-arrestin protein family. This protein binds to Trx and it blocks the antioxidative function of thioredoxin permitting the accumulation of ROS and cellular stress. TxNIP is induced in response to cellular stress, and it is associated to hyperglycemia and other stress diseases. It has been proposed that under normal conditions, endothelial cells contain a significant level of Trx activity due to the downregulation of TxNIP [36], but hyperglycemia-induced vascular dysfunction induces that protein.

#### 5.1.5. Paraoxonases (PON)

Paraoxonases are another family of proteins involved in the antioxidant mechanisms against oxidative stress. The human family consists of three members encoded by three different genes, PON1, PON2 and PON3 [214]. They do not belong to the set of intracellular antioxidant enzymes regulated by the Nrf2 –ARE pathway. Paraoxonases are elusive blood proteins with esterase but also oxidoreductase activity, mostly associated to HDL, and they show the capability to protect several plasmatic lipoproteins against lipid oxidation blocking the formation of lipoprotein-oxidized forms, especially oxidized LDL. Although they do not directly extinguish ROS, they contribute significantly to the redox balance in the plasma and decrease cardiovascular risk. In addition to the extracellular location, it has been described that the PON2 isoenzyme is able to bind coenzyme Q_10_ in mitochondrial complex III, partially reducing mitochondrial ROS production [215]. However, the intracellular role of paraoxonases is uncertain due to their somewhat elusive nature in comparison to that of other types of antioxidant enzymes. It was proven that its activity is decreased in hyperglycemia and aged patients [216,217] and therefore paraoxonase activity has clinical importance and is used as biochemical marker of the blood redox index. In addition, some bioactive molecules with beneficial effects on hyperglycemia or vascular diseases, including dietary polyphenols or acetylsalicylate, stimulate PON1 activity [218,219]. Recent reviews of the role of paraoxonases have been published [220], but further research is needed to clarify the nature and mechanisms of action and the authentic importance of these proteins in the maintenance of the redox equilibrium.

#### 5.1.6. Sirtuins (Sirt)

Sirtuins are the last proteins included in this review that potentially could be used for preventing hyperglycemia and the associated oxidative stress as they are involved in the modulation of these conditions even though they are not part of the Nrf2-mediated antioxidant response. This family of proteins are NAD^+^-dependent protein deacetylases [221,222]. Currently, the family consist of seven members [223] distributed in the cellular compartments. Sirt1 and Sirt2 are located in the cytoplasm and nucleus as they can undergo translocation, Sirt3 is found in the mitochondrial matrix, and Sirt6 is located only in the nucleus. The first studies on the importance of protein deacetylation were focused on nuclear histones and chromatin regulation for cell cycle and survival [224]. Sirtuins were promptly associated with an antiaging action in that they prolong lifespan [225]. As is well-known, calorie restriction can retard the aging process and delay the onset of numerous age-related disorders, including mammalian DMT2 or just the development of insulin resistance. It has been demonstrated that each caloric restriction induces histone deacetylation and that Sirt activation mimics caloric restriction [226,227]. Oppositely, aging and hypercaloric diet inhibit sirtuins and enhance hyperglycemia. Accordingly, sirtuins were proposed as important players in the pathogenesis of DMT2 [228].

In addition to histones, the number of proteins regulated via sirtuin-dependent deacetylation has been greatly increased, including the transcription factors NFκB [229], Nrf2 [230,231] and FoxO [190,232,233]. All of them are related to the induction or response to oxidative stress as previously discussed. Furthermore, related to glucose metabolism and sirtuin action, any decrease in NAD^+^ would hinder the deacetylating action of these proteins according to the sirtuin cofactor, leading to higher levels of over-acetylated proteins. This modification produces remarkable functional changes in the metabolism. Thus, the a NADH/NAD^+^ ratio induced via the activation of the polyol pathway due to hyperglycemia will attenuate sirtuin activity [230]. According to this, numerous studies have demonstrated the attenuated expression of sirtuins in diabetes. Reciprocally, the decrease in the cellular content of sirtuins leads to oxidative stress, less efficient glucose metabolism and diabetic complications [234], emphasizing the multiple interactions between hyperglycemia and oxidative stress. Several studies have proposed that in addition to caloric restriction, supplementation with nicotinamide riboside as a NAD^+^ precursor can serve as an approach for enhancing sirtuin activity thereby augmenting protein deacetylation and preventing hyperglycemia and its pathological complications. For instance, using cultured cells in the presence of high glucose content, supplementation of the media with nicotinamide riboside induced Sirt3, GPx1 and SOD2 and cell viability [235], suggesting that Sirt3 may play a role in increasing antioxidant defense to provide resistance against hyperglycemia-associated oxidative stress. Sirtuin activation has been recently included as a suitable approach in the therapeutical treatment of diabetic nephropathy [236] and retinopathy [237].

## 6. Concluding Remarks and Perspectives

Undoubtedly, carbohydrates are essential nutrients for human metabolism, and glucose is the main structural unit to be metabolized mostly under aerobic conditions. On the other hand, the redox equilibrium is a keystone in human health. The use of oxygen for metabolic oxidations is a permanent risk for that equilibrium, as oxygen is a source of ROS. Low amounts of ROS are important messengers that regulate signals related to various biological processes, such as cell growth, differentiation, senescence, apoptosis, and autophagy. However, when ROS production exceeds the buffering capacity of the antioxidant defense systems or when antioxidant enzymes are defective, oxidative stress occurs, leading to cellular dysfunction, cell death, and in the last stage whole organ failure.

Hyperglycemia exacerbates the production of both mitochondrial and cytosolic ROS, and the subsequent oxidative stress contributes to the appearance of a variety of pathological complications related to diabetes. However, despite the thousands of studies, our integral understanding of molecular events during oxidative stress under chronic hyperglycemia is limited so far. This is probably due to the number of different processes and factors involved in hyperglycemia-induced pathological complications. Hyperglycemia and oxidative stress show multiple self-stimulatory interconnections. Moderate inflammation can be considered the third feature of the vicious triangle.

Hyperglycemia induces mitochondrial damage, collateral glucose routes and spontaneous glucose reactions. All processes promote an excessive ROS generation leading to oxidative stress. Looking in the opposite direction, oxidative stress induces the AGE–RAGE pathway, which finally stimulates further mitochondrial impairment, glucose collateral routes, and insulin resistance, leading to hyperglycemia reinforcement. The multiple crosstalks among the pathways make it difficult to establish the relative contributions of each one to the global level of oxidative stress. The particular contributions to the tissue-specific harmful effects also remain an outstanding but unresolved question.

Fortunately for the cells, sometimes, the crosstalks among ROS-generating glucose routes are not self-stimulatory, but counteract each other and contribute to ameliorating hyperglycemia. For instance, as the NADPH level lowers, the antioxidant response due to the GPx/GR pair and reduced glutathione is weakened, but the contribution of NOX action to generating superoxide and oxidative stress also decreases.

The pro-oxidant and the opposite antioxidant responses can be focused on the main transcription factors that regulate both processes. This view could be (over)simplified as a fight between NFκB and Nrf2 with a certain advantage for the first one due to the predominance of glucose aerobic metabolism. NFκB is the transcription factor mainly involved in collecting pro-oxidant signals and a keystone in the reciprocal stimulatory action between hyperglycemia and oxidative stress. NFκB enhances the expression of molecular modulators related to the tissue-specific pathological complications associated to retinopathy, nephropathy or vascular dysfunction. On the other hand, Nrf2 is the major factor triggering the antioxidant response to break the vicious cycle between oxidative stress and hyperglycemia.

From the redox point of view, NFκB is able to induce a series of enzymes able to generate ROS, and Nrf2 is able to induce a series of antioxidant enzymes and accessory proteins able to scavenge ROS. The interconnections among the distinct pro-oxidant enzymes are also common. In addition to the mitochondrial ETC, NOX is perhaps the most important enzyme generating superoxide. NOX is also a suitable enzyme with which to illustrate the dual role of ROS as beneficial or harmful agents. NOX-derived ROS allow immune system cells to promote chemotaxis and inflammation to permeabilize the endothelium and eventually neutralize pathogens in infections via massive ROS production, but ROS-derived NOX overexpression is involved in endothelium dysfunction. NOX increase, mostly an increase in the NOX4 isoenzyme, can determine the activation of other pro-oxidant enzymes such as NOS or XO in a tissue-dependent fashion. Those interconnections make difficult the counteraction of the antioxidant response, allowing hyperglycemia to easily progress and diabetes pathological complications to proceed.

Concerning the perspectives, the limited current knowledge about the multiple interconnections and reciprocal self-stimulatory mechanisms between hyperglycemia and oxidative stress accounts for the great difficulty in finding effective treatments, but also provides several different approaches for at least ameliorating their pathological complications. The external regulation of the NFκB, Nrf2 or FoxO master transcription factors involved in the redox equilibrium is difficult. However, NFκB expression can be diminished via AMPK activation. In practice, this means low caloric ingestion with moderate to low glucose ingestion and controlling the glycemic index to minimize ROS generation via glucose metabolism. Nevertheless, mitochondrial respiration is prominent in all conditions. Enhancers of mitohormesis would be effective. PKC action or spontaneous glucose fragmentation into methylglyoxal and other small ketoaldehydes activate the AGE–RAGE pathway, which seems an effective way to promote the expression of NFκB and its pro-oxidant reinforcement. The polyol pathway seems to be quantitatively significant, as up to 30% of glucose can be metabolized via this route, and this is a suitable target. The decreases in GSH content due to decreased levels of NADPH in sirtuin caused by lowering levels of NAD^+^ and the fructose-derived AGEs seem to also point to the important contribution of the polyol route to oxidative stress and hyperglycemia reinforcement. Treating with aldose reductase inhibitors is a possible effective approach to inhibit this pathway.

Focusing on approaches to increase Nrf2 expression, some antioxidant micronutrient supplements are effective at inducing this transcription factor, and they could be appropriate for inducing a powerful and durable antioxidant response. These antioxidant biomolecules should be able to be easily absorbed into the blood and inside cells for inducing Nrf2 expression, but they should also have the capability to be direct ROS scavengers. Agents that induce native FoxO but do not enhance post-translational modifications that change the final effect from an antioxidant one to a pro-oxidant one are difficult to obtain, so this transcription factor is not a suitable target for therapeutical approaches, at least not directly.

In addition to the possible effects on master transcription factors, particular approaches to minimize oxidative stress are also possible. Tetrahydrobiopterin is needed for maintaining eNOS as a dimer for NO production, avoiding peroxynitrite generation and vascular dysfunction. Other specific agents related to the inhibition of TGFβ have been proposed, such as artemisinin. The tissue dependence of the action could be considered according to the relative importance of the molecular factors in the damaging effects. In this regard, the polyol pathway seems to be more important in retinopathy, TGFβ synthesis seems to be more important in nephropathy and NO impairment seems to be more important in the vascular endothelium. The inhibition of aldose reductase, or artemisinin or BH_4_ supplementation could be plausible approaches for the amelioration of those diabetes complications. Others, such as AGE inhibitors or scavengers, nicotinamide riboside supplementation for sirtuin deacetylating activity or appropriate supplementation of sulfur- and selenium-containing supplements for GPx/GR and Trx/TrxR pair activities to maintain appropriate glutathione and Trx levels seem to be but valid approaches for antioxidative therapy. Perspectives should be focused on the use of a mixture of agents targeting different glucose pathways or favoring Nrf2 but counteracting NfκB expressions. Of course, co-ordinated and combinatory actions seem to be crucial.

## Data Availability

Data sharing is not applicable to this article. No new data were created for writing the review except in the designing of the figures. All published material is appropriately referenced.

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
