# Peer review of "Hyperglycemia and Oxidative Stress: An Integral, Updated and Critical Overview of Their Metabolic Interconnections"

_ijms, 2023, doi:10.3390/ijms24119352_

Round 1

Reviewer 1 Report

Dear Authors,

I have read your manuscript entitled "Hyperglycemia and oxidative stress: An integral, updated and critical overview on their metabolic interconnexions". Congratulations for your work. It has been written in a clear and descriptive manner.

Kind regards

Dear Sirs/Madams,

I have read your manuscript entitled "Hyperglycemia and oxidative stress: An integral, updated and critical overview on their metabolic interconnexions".

The whole work has been written in a clear and descriptive manner.

I have identified some minor typos and signed them on your original pdf.

- There are too many hyphenation problems.

- There are some abbreviations that they weren’t explained when they first used. So this needs correction.

- I found a wrong reference style in the text.

- I found a few wrong symbols like gamma and omega

I recommend you read the whole text and correct those typos.

Author Response

First of all, point.by-point replies are written in red inserted between to your original comments. The manuscript has been amended according to those replies.

Dear Sirs/Madams,

I have read your manuscript entitled "Hyperglycemia and oxidative stress: An integral, updated and critical overview on their metabolic interconnexions". The whole work has been written in a clear and descriptive manner.

Thank you very much for your review and your introductory comments.

I have identified some minor typos and signed them on your original pdf.

- There are too many hyphenation problems.

Right. We are so sorry about that. We have carefully revised the manuscript and the English has been edited. According to that, numerous hypos have been corrected. They are marked in red in the amended version of the manuscript. Hyphenations were automatically introduced by the Microsoft program. I have eliminated this function in the revised version.

- There are some abbreviations that they weren’t explained when they first used. So this needs correction.

Thank you. All abbreviations have been described with the full names the first time that they appear. Some of them are anecdotal names which not really facilitate comprehension, but we cannot go further explaining the historical reasons of those names.

- I found a wrong reference style in the text.

The style of the references was according to the instructions to authors. They are numbered and inserted in brackets by order of appearance. A couple of wrong styles have been detected during the careful editing and subsequently corrected. Please note renumbering because five references have been added according to suggestions from reviewer 3.

- I found a few wrong symbols like gamma and omega

Thank you. We have found one g by gamma at Table 2 and one w by omega referred to omega3 fatty acids in the text. Both letters have been replaced by the Greek g and w. In addition, e has been replaced by e in the paragraph related to protein glycation on the e-amino group of Lysine residues.

I recommend you read the whole text and correct those typos.

Of course, That is what we did. Thank you very much for your help and time devoted to this review.

Warmest regards

Reviewer 2 Report

In the manuscript entitled ... the authors presented the latest scientific findings related to the interrelationship of oxidative stress processes that occur as a consequence of increased glycemia and the action of the antioxidant system, all in the light of the development of associated symptoms of metabolic disorders.

The review was conceived clearly and without redundant topics. In this way, readers can easily follow the logical course of the text's development. The authors of the manuscript begin by describing the process of oxidative stress that is a consequence of increased glycemia, and state the ways in which such stress can lead to pathological changes. Then, at the molecular level, they describe alternative glucose metabolic pathways that contribute to the development of oxidative stress to the greatest extent. Logically, after that they also describe the mechanisms of antioxidant defense of cells and the organism in order to finally draw conclusions about the importance of actively maintaining the balance between pro-oxidative and antioxidant processes in preventing the development of metabolic diseases. The authors bring all this into direct connection with modern approaches in the treatment of patients who have already suffered from DMT, but also with a preventive approach that can be applied by all healthy people.

The literature used is contemporary, although it is noticed that for some statements, references from the time when the phenomenon was originally described (30 or 40 years ago) were used.

Taking everything into account, I believe that it is an interesting magazine that presents contemporary knowledge in the given field in an orderly and logical manner and that will be interesting to many readers, both those directly involved in the given field and all those readers who possess basic knowledge and they want to know more.

However, while reading the manuscript, I came across some details that I would like to draw the authors' attention to, so they are listed in the text below as a minor revision:

Line 89: from the Scheme1 it cannot be seen what refers to (a) or (b)

Line 90: HavBer-Weiss

Line 91: …should also BE considered

Line 106: …any ROS is harmful for cells AND that any antioxidant…

Line 133: that that

Line 222: The above-mentioned pathways as WERE summarized in Table 1.

Line 238: 2.

Line 229: acetyl-CoA – this word appeared for the first time in line 65 as AcCoA. I believe that it would be more appropriate to named it as Acetyl-CoA in the line 65 (introducing the abbreviation) and then as AcCoA throughout the text.

Line 373: reduced_GSH

Line 616: nucleotide-binging; nucleotide-binDing?

Line 845: the full stop is missing

Line 900: Both, the Nrf2 and GLUT4 transportedR

Line 948: also involveD

Line 957: several signal transduction pathwayS

Line 973: Abbreviation SOD reffers to the singular named in the subtitle above (Line 972) and jet, there is stated that SOD are enzymes (plural). That’s why I recommend to write it as SODs.

Line 974, 994, 1000, 1001, 1005, 1008, 1010, 1023, 1029: H2O2 should be written as alsewhere (2 in subscript).

Line 1010: This is a selenium-protein that catalyzes the disappearance of H2O2 coupled to the glutathione oxidation. I would say that this is an missleading word because it suggests the disappearance of the molecule instead of its conversion into something else (water).

Line 1030: Prx6 has only one Cys residue at the catalytic active site that activity that is oxidized to sulfenic acid by the H2O2 – I believe that this part of the sentence should be rewritten since I am not sure if I can understand it properly.

Line 1160: Nrf2 in IS the major factor…

Line 1205: TFG into TGF

Author Response

First of all, point.by-point replies are written in red inserted between to your original comments. The manuscript has been amended according to those replies.

Comments and Suggestions for Authors

In the manuscript entitled ... the authors presented the latest scientific findings related to the interrelationship of oxidative stress processes that occur as a consequence of increased glycemia and the action of the antioxidant system, all in the light of the development of associated symptoms of metabolic disorders.

The review was conceived clearly and without redundant topics. In this way, readers can easily follow the logical course of the text's development. The authors of the manuscript begin by describing the process of oxidative stress that is a consequence of increased glycemia, and state the ways in which such stress can lead to pathological changes. Then, at the molecular level, they describe alternative glucose metabolic pathways that contribute to the development of oxidative stress to the greatest extent. Logically, after that they also describe the mechanisms of antioxidant defense of cells and the organism in order to finally draw conclusions about the importance of actively maintaining the balance between pro-oxidative and antioxidant processes in preventing the development of metabolic diseases. The authors bring all this into direct connection with modern approaches in the treatment of patients who have already suffered from DMT, but also with a preventive approach that can be applied by all healthy people.

The literature used is contemporary, although it is noticed that for some statements, references from the time when the phenomenon was originally described (30 or 40 years ago) were used.

Taking everything into account, I believe that it is an interesting magazine that presents contemporary knowledge in the given field in an orderly and logical manner and that will be interesting to many readers, both those directly involved in the given field and all those readers who possess basic knowledge and they want to know more.

First of all, thank you very much for your review, your introductory comments, and your careful revision of the manuscript. We are very grateful. Concerning the point related to the literature cited in this review, we have tried to cite new data for writing an updated review, but as you emphasize, we have also introduced old references that were cornerstone for the advance of the field. Old authors deserve somehow a tribute and we have tried a small contribution to that idea. I mean for instance the references related to glutathione role as antioxidant due to the glutathione peroxidase/glutathione reductase cycle, or references about the role of NADPH and glucose-6-phosphate dehydrogenase in fabism (ref. 128,129, one pioneer work and one very recent review). Pioneer papers related to the main transcription factors mentioned (mostly NFkB and Nrf2) are also referenced as long as recent advances. 

However, while reading the manuscript, I came across some details that I would like to draw the authors' attention to, so they are listed in the text below as a minor revision:

We are so sorry about that. We have carefully revised the manuscript and the English has been edited. According to that, all hypos that you listed have been corrected. They are marked in red in the amended version of the manuscript. Please  note that sometimes the lines have been shifted due to other minor modifications suggested by the other reviewers.

Line 89: from the Scheme1 it cannot be seen what refers to (a) or (b)  Corrected.

Line 90: HavBer-Weiss  Corrected.

Line 91: …should also BE considered . Added.

Line 106: …any ROS is harmful for cells AND that any antioxidant…Added.

Line 133: that that                                            Duplication has been deleted.

Line 222: The above-mentioned pathways as WERE summarized in Table 1. Corrected.

Line 238: 2. Puntuation has been added.

Line 229: acetyl-CoA – this word appeared for the first time in line 65 as AcCoA. I believe that it would be more appropriate to named it as Acetyl-CoA in the line 65 (introducing the abbreviation) and then as AcCoA throughout the text. Thank you for your suggestion. We did exactly that for this and other abbreviations.

Line 373: reduced_GSH. Corrected.

Line 616: nucleotide-binging; nucleotide-binDing? Corrected.

Line 845: the full stop is missing. Added.

Line 900: Both, the Nrf2 and GLUT4 transportedR . Corrected.

Line 948: also involved. Added.

Line 957: several signal transduction pathwayS. Added.

Line 973: Abbreviation SOD reffers to the singular named in the subtitle above (Line 972) and jet, there is stated that SOD are enzymes (plural). That’s why I recommend to write it as SODs. You are right. We did it

Line 974, 994, 1000, 1001, 1005, 1008, 1010, 1023, 1029: H2O2 should be written as alsewhere (2 in subscript). Of course. All these lines and some other have been corrected.

Line 1010: This is a selenium-protein that catalyzes the disappearance of H2O2 coupled to the glutathione oxidation. I would say that this is an missleading word because it suggests the disappearance of the molecule instead of its conversion into something else (water). We understand. You are right again. The sentence has been re-written (line 1037 in the amended version).

Line 1030: Prx6 has only one Cys residue at the catalytic active site that activity that is oxidized to sulfenic acid by the H2O2 – I believe that this part of the sentence should be rewritten since I am not sure if I can understand it properly. Right. The sentence has been re-written (lines 1056-1059)

Line 1160: Nrf2 in IS the major factor… Corrected

Line 1205: TFG into TGF. Corrected

Warmest regards

Submission Date                09 May 2023

Date of this review              21 May 2023 10:15:53

Date of the amended version: 24 May 2023

Reviewer 3 Report

In this manuscript authors summarized the key factors linking the diverse glucose metabolic routes enhanced in hyperglycemia with ROS formation and vice versa.

The manuscript is interesting and well illustrated. Only minor points deserve to be improved. In particular:

Lines 843-852: Since this is a review article, the multicafeted role of NRF2/KEAP1 signalling deserves to be mentioned. In fact, this signalling is also involved in the progression and onset of several types of cancer (as recently reviewed PMID: 35901941, 36335520, 36641100)

Lines 180-184: It deserves to be specified that the altered ECM balance in OA is mainly due to the increased levels of proteases that degrade ECM components (see  PMID: 35131488, 36676051 )

References must be written in full length when mentioned for the first time

An accurate revision of reference in the main text is recommended

Author Response

Reply to reviewer 3.

First of all, point.by-point replies are written in red inserted between to your original comments. The manuscript has been amended according to those replies.

Comments and Suggestions for Authors

In this manuscript authors summarized the key factors linking the diverse glucose metabolic routes enhanced in hyperglycemia with ROS formation and vice versa.

The manuscript is interesting and well illustrated. Only minor points deserve to be improved. In particular:

Thank you very much for your review and your introductory comments. Point.by-point replies to your specific comments are written in red. The manuscript has been amended according to those replies.

Lines 843-852: Since this is a review article, the multicafeted role of NRF2/KEAP1 signalling deserves to be mentioned. In fact, this signalling is also involved in the progression and onset of several types of cancer (as recently reviewed PMID: 35901941, 36335520, 36641100).

You are right concerning the important roles of Nrf2 in other pathological processes in addition to the antioxidant response. We have now added a few lines concerning to the role in the progression of some types of cancer (lines 863-869) in the revised version). Your suggested references have been added and numbered 172-174, as follows:

172.Tossetta G, Marzioni D. Natural and synthetic compounds in Ovarian Cancer: A focus on NRF2/KEAP1 pathway. Pharmacol Res. 2022, 183, 106365, doi:10.1016/j.phrs.2022.106365.

  1. Marzioni, D., Mazzucchelli, R., Fantone, S. & Tossetta, G. NRF2 modulation in TRAMP mice: an in vivo model of prostate cancer. Mol Biol Rep. 2023, 50(1), 873-881, doi:10.1007/s11033-022-08052-2.
  2. Tossetta, G. & Marzioni, D. Targeting the NRF2/KEAP1 pathway in cervical and endometrial cancers. Eur J Pharmacol. 2023, 941, 175503, doi:10.1016/j.ejphar.2023.175503.

Lines 180-184: It deserves to be specified that the altered ECM balance in OA is mainly due to the increased levels of proteases that degrade ECM components (see PMID: 35131488, 36676051 )

We agree with you again concerning the essential role of metalloproteinases in OA. Oxidative stress in osteocartilaginous tisse enhances expression of metalloproteinases. A brief sentence has been added concerning this issue (lines 184-187). Appropriate references have been also added to mention this point as follows:

43.Ashruf, O.S. & Ansari, M.Y. Natural Compounds: Potential Therapeutics for the Inhibition of Cartilage Matrix Degradation in Osteoarthritis. Life 2023, 13, 102, doi.org/10.3390/life13010102

44.Tossetta, G. Fantone, S., Licini, C., Marzioni, D. & Mattioli-Belmonte. M. The multifaced role of HtrA1 in the development of joint and skeletal disorders. Bone, 2022, 157116350, doi.org/10.1016/j.bone.2022.116350.

References must be written in full length when mentioned for the first time

We have written the full names of all abbreviations the first time they appear in the text. Some of them are anecdotal names which not really facilitate comprehension, but we cannot go further explaining the historical reasons of those names.

An accurate revision of reference in the main text is recommended

The revision concerning references and cross-references to the different sections has been very careful. We found a couple of mistakes, that have been corrected.

Warmest regards